# Marine-Derived Bioactive Peptides Self-Assembled Multifunctional Materials: Antioxidant and Wound Healing

**DOI:** 10.3390/antiox12061190

**Published:** 2023-05-31

**Authors:** Dingyi Yu, Shenghao Cui, Liqi Chen, Shuang Zheng, Di Zhao, Xinyu Yin, Faming Yang, Jingdi Chen

**Affiliations:** 1Marine College, Shandong University, Weihai 264209, China; 202000810156@mail.sdu.edu.cn (D.Y.); 202000810226@mail.sdu.edu.cn (S.C.); 201900810011@mail.sdu.edu.cn (L.C.); 202100810142@mail.sdu.edu.cn (S.Z.); 201900810068@mail.sdu.edu.cn (D.Z.); 202100810117@mail.sdu.edu.cn (X.Y.); 2Shandong Laboratory of Advanced Materials and Green Manufacturing, Yantai 265599, China

**Keywords:** marine-derived bioactive peptides, self-assembled peptides materials, wound healing, topical application, molecular docking

## Abstract

Peptide self-assembling materials have received significant attention from researchers in recent years, emerging as a popular field in biological, environmental, medical, and other new materials studies. In this study, we utilized controllable enzymatic hydrolysis technology (animal proteases) to obtain supramolecular peptide self-assembling materials (CAPs) from the Pacific oyster (*Crassostrea gigas*). We conducted physicochemical analyses to explore the pro-healing mechanisms of CAPs on skin wounds in both in vitro and in vivo experiments through a topical application. The results demonstrated that CAPs exhibit a pH-responsive behavior for self-assembly and consist of peptides ranging from 550 to 2300 Da in molecular weight, with peptide chain lengths of mainly 11–16 amino acids. In vitro experiments indicated that CAPs display a procoagulant effect, free radical scavenging activity, and promote the proliferation of HaCaTs (112.74% and 127.61%). Moreover, our in vivo experiments demonstrated that CAPs possess the ability to mitigate inflammation, boost fibroblast proliferation, and promote revascularization, which accelerates the epithelialization process. Consequently, a balanced collagen I/III ratio in the repaired tissue and the promotion of hair follicle regeneration were observed. With these remarkable findings, CAPs can be regarded as a natural and secure treatment option with high efficacy for skin wound healing. The potential of CAPs to be further developed for traceless skin wound healing is an exciting area for future research and development.

## 1. Introduction

The skin has a complex structure and skin wound healing undergoes four overlapping stages: hemostasis, inflammation, proliferation, and remodeling [1,2,3]. When skin wounds fail to heal in an orderly and timely manner, it often results in chronic wounds. Such wounds may present with various complications, including poor circulation, hypoxia, bacterial infections, elevated levels of white blood cells, and so on. These issues can exacerbate the healing process and prolong the recovery time [4,5]. The ultimate goal for both physicians and scientists are the perfect regeneration of wound tissue and scarless repair, including the epidermis, dermis, and accessory organs. With intensive research into the role and mechanisms of various microenvironments in wound healing, therapeutic options have been proposed to address and modulate the biological, biochemical, and physical-pathological microenvironments of wounds. New biomaterials are currently in fashion. Unfortunately, clinically challenging biomaterials that address and pro-mote skin regeneration and prevent the formation of scar tissue have rarely been investigated. Second, the biomaterial manufacturing process is most complex and expensive and unsuitable for industrial fabrication. Moreover, it is functionally difficult to meet the demands of complex healing processes [6,7]. Therefore, it is necessary to develop multifunctional, safe, efficient, and low-cost drugs and treatment strategies.

Marine-derived bioingredients have garnered significant interest due to their distinctive structures and biomedical potential, which are a direct result of the extreme physicochemical properties of the marine environment. Meanwhile, these compounds possess novel therapeutic properties and offer several advantages over traditional remedies. As a result, the exploration of marine resources has become increasingly popular in the biomedical research community [8,9]. The *Crassostrea gigas* [10] are an important aquaculture shellfish along the Pacific coast of Asia and are heavily underutilized. In recent years, *Crassostrea gigas* are characterized by the potential of multifunctional bioactivities, including antioxidation, antimicrobial, antitumor, etc. [11,12,13], which lays a theoretical foundation for this study of the effect on soft tissue repair. Among them, enzymatic preparation of active peptides is not only an important way to achieve their high-value utilization but also an important means to explore the related functional activities. For example, an anticoagulant heptapeptide (P-3-CG) was obtained by hydrolysis of *Crassostrea gigas* flash using pepsin [10]. As a result, they are valuable as commodities and feedstocks in the aquaculture and food industries, while their medical value has not been exploited.

Peptides can convert into various nanostructures through self-assembly and exhibit interesting features including mechanical stability, thermal stability, as well as semiconductor, piezoelectric, and optical properties [14,15]. Most of the peptides currently used in this study of self-assembled materials are synthetic or purified peptides. While this is advantageous for modulating material morphology, it comes with drawbacks such as high cost and low yield. Controlled enzymatic digestion techniques, which can yield enzymatic digests rich in peptides, amino acids, and metal ions, can help overcome these issues. However, these techniques present challenges in preparing self-assembled materials with controlled morphology. J.P. Wu, a renowned scholar in the food field, has suggested that purified fractions obtained from enzymatic digests using controlled pressurized membrane separation equipment may be a promising alternative for peptide self-assembly studies [16]. Meanwhile, the obtained random peptide assemblies glatiramoids and (KEYA) n Q11 can be prepared by a random polymerization of amino acids [17]. Hence, it will be an effective and feasible approach to propose a new strategy for preparing peptide self-assembling materials by using the interdisciplinary intersection.

To create a multi-functional supramolecular material of peptides with randomized self-assembling properties, we were inspired by the chemical one-pot synthesis method [18] and the strategy of active peptide self-assemblies [17]. Our study improved the peptide preparation method by employing a controlled hydrolysis technique and animal proteases to extract the material from the flash of *Crassostrea gigas*. After physicochemical analysis, we found that it has self-assembly properties and is pH-responsive, with multiple peptide chains self-assembling to form spheres under a variety of pH environments. To verify the healing activity of the extracted materials, we conducted activity screening using an in vitro system. We then investigated the healing mechanism through the topical application of the material on a mouse model of trauma. Furthermore, to verify its safety and biocompatibility, we examined visceral pathological sections.

## 2. Materials and Methods

### 2.1. Materials

*Crassostrea gigas* flash was sourced from the Imperial Palace Comprehensive Market in Weihai City, Shandong in December 2022, and Yunnan Baiyao, which is a product made from Chinese medicinal materials containing various active ingredients with a variety of wound healing properties was sourced from Yunnan Baiyao Group Co., Ltd. (Kunming, China). Animal experiments were performed using male SPF mice weighing 20 ± 2 g (sourced from Sibei Fu (Beijing) Biotechnology Co., Ltd., Beijing, China). The company possesses a production license (No. SCXK (Jing) 20190010). Pangbo Biological Engineering Co., Ltd. (Xintai, China). provided animal protease (3 × 10^5^ U·g^−1^). The Total Antioxidant Capacity Test Kit (FRAP) was acquired from Beyotime Biotechnology Co., Ltd. in Shanghai, China. The Cell Counting Kit-8 (CCK-8) was obtained from APE x BIO, a company based in Houston, TX, USA. Finally, the AO/EB Staining Kit was purchased from Sangon Biotech (Shanghai) Co., Ltd., which is also based in Shanghai, China.

### 2.2. Preparation of Randomized Self-Assembling Materials (CAPs)

Preparation of CAPs was by controlled enzymatic enrichment of peptides [19]: Freshly washed *Crassostrea gigas* were homogenized by high-speed shear, and 1000 U/g of protease was added to the homogenate and enzymatically digested for 3–6 h. After enzymatic digestion, the enzyme was inactivated at 100 °C for 10 min; the fractions with molecular weights less than 3000 Da were separated by ultrafiltration membranes (GC-UF0031 made of PES, GE, USA) under refrigerated temperature conditions. The fractions with molecular weights less than 3000 Da were spun concentrated (N-1300V, EYELA, Tokyo, Japan), freeze-dried (FDU-2110, EYELA, Japan) into powder, and set aside.

Then, dissolved in PBS buffer at 50–200 mg/mL, pH adjusted to 2.0, 7.4, 8.4, 9.4, 10.4 with a pH adjuster, vortexed self-assembly at room temperature for 30 min, followed by low-speed centrifugation to discard the supernatant and the precipitate was freeze-dried into powder to obtain the described CAPs.

### 2.3. Physical and Chemical Analysis of CAPs

Molecular weight distribution and major peptide sequences were first analyzed using liquid chromatography-mass spectrometry (LC-MS/MS).

Afterwards, the particle size and zeta of 50 mg/mL and 200 mg/mL samples were measured by a dynamic laser scattering (DLS) method at 25 °C using a Zetasizer Nano ZS instrument (Malvern Instruments Ltd., Worcestershire, UK). All measurements below were performed in triplicate.

FTIR spectroscopy was performed using an IRSpirit-1 instrument (Shimadzu, Japan) with a resolution of 4 cm^−1^ between 4000 cm^−1^ and 400 cm^−1^. Background measurements were performed using blank KBr plates.

Samples of CAPs at concentrations of 50 and 200 mg/mL in PBS solutions at pH 2.0, pH 7.4, pH 8.4, pH 9.4, and pH 10.4 were observed in the sample observation room using SEM (Nova Nano SEM 450, FEI, Hillsboro, OR, USA).

Afterwards, ultrafiltered, purified, and concentrated lyophilized powder was used as the control group to compare the self-assembly effect. In the same way, the effects of ionic strength and temperature on their structures were explored sequentially. The CAPs were diluted to 1 mM with deionized water and then transferred to a 0.1 cm quartz cuvette. CD spectra from 190 to 260 nm at 25 °C were recorded using a circular dichroism spectrometer (Jasco J-1500, Shanghai, China).

Samples were dispersed in deionized water, and a small amount was spotted on a clean mica sheet and then air-dried. The morphology of the samples was scanned with an atomic force microscope (AFM, JPK NanoWizard 4, Berlin, Germany).

The physicochemical analysis of CAPs was carried out according to previous reports and the methods are detailed in the Appendix A [20].

### 2.4. Verification System of In Vitro Healing Activity

#### 2.4.1. Test of Antioxidant Activity

The test of antioxidant activity of the CAPs was carried out according to previous reports and the methods are detailed in the Appendix A [21,22].

#### 2.4.2. The Test of In Vitro Plasma Recalcification Time

To obtain plasma, fresh rabbit hematology was anticoagulated and centrifuged at 1000 rpm for 10 min. In each tube, we added 0.1 mL of plasma and 0.1 mL of CAPs samples. Yunnan Baiyao (1 mg/mL) was used as the positive control, while a 2% citric acid solution was used as the negative control. We then incubated the mixture at 37 °C for 1 min. Next, we added 0.1 mL of a calcium chloride solution (0.025 mol/L) to each tube and immediately started the stopwatch. The tube was swirled every 15 s until a white granular fibrin appeared, and the time was recorded. We performed each test 5 times.

#### 2.4.3. Cell Proliferation and Toxicity Test

The cell proliferation and toxicity test were carried out according to previous reports and the methods are detailed in the Appendix A [19].

### 2.5. Animal Grouping and Experiment

Newly purchased mice were randomly allocated to 3 groups of 9 mice each: negative control group (control), positive control group (Yunnan Baiyao, YN), and CAPs-treated group (CAPs). CAPs, which had a pH range of 6.06–5.93 and were prepared using deionized water, were administered topically as a paste at a dosage of 3–5 mg/day. Prior to surgery, the mice were housed under standard laboratory conditions for one week. On the day of the experiment, the mice were anesthetized with a 1% intraperitoneal injection of pentobarbital sodium (50 mg/kg) and a full-thickness skin wound model was created on the dorsal skin, measuring 0.8 cm in diameter. The groups (YN and CAPs) received their respective treatments, while the control group was left untreated. The size of the wound was measured every 2 days and photographs were taken. The experiments were conducted in triplicate to ensure the precision and consistency of the results.

### 2.6. Healing Rate

After inducing the skin wound model, the wounds were photographed every second day using the Canon camera (EOS 850D, Japan). Image J software (the National Institutes of Health in Bethesda, Bethesda, MD, USA) was used to estimate the wound healing rate. Based on the following wound healing rate calculation:$$Wound\;healing\left(\%\right)=(S_{0}-S_{n})/S_{0}\times 100$$
where $S_{0}$ is the diameter of the wound at modeling and $S_{n}$ is the diameter of the wound on the designated timepoint.

### 2.7. Histology

For histological assessment, the groups of mice were sacrificed on days 3, 7, and 14 after modeling and sections of skin tissue were collected and fixed in 4% paraformaldehyde, which was then sliced into 5 μm thickness. On the 14th day, the mice’s kidneys, spleen and liver were collected and sectioned at the same time. The sections were stained with hematoxylin-eosin (H&E) and Masson’s staining, and observed under a microscope (Axio Observer, ZEISS, Oberkochen, Germany).

### 2.8. Immunohistochemical Method

The levels of platelet endothelial cell adhesion molecule-1 (PECAM-1, CD31) and fibroblast growth factor-10 (FGF-10) were measured using immunohistochemistry. Tissue sections were harvested and subjected to dewaxing, followed by antigen retrieval in distilled water. Endogenous peroxidase was blocked using a 3% H_2_O_2_-methanol solution, and the antigen was retrieved with a citrate buffer (pH 6.0). After that, the sections were incubated with a primary antibody that had been diluted in a humidified atmosphere and left to incubate at 37 °C for 2 h. After sealing, the sections were washed with PBS and incubated at room temperature with a universal IgG antibody-Fab fragment-HRP multimer. Finally, the sections were analyzed using Image-pro plus 6.0 software and viewed under an optical microscope (Axio Observer, ZEISS, Oberkochen, Germany).

### 2.9. Sirius Red Picric Acid Staining

The Sirius Red Picric Acid Dyeing was carried out according to the manufacturer’s protocol. Initially, graded ethanol was used to rehydrate the sections after dewaxing. Following that, a picrosirius red dye was applied to the sections for 8–10 min. Next, the sections were rinsed with running water and put into absolute ethanol to remove water and mounted with neutral balsam. Slides were observed with a microscope (Eclipse Ci-L, Nikon, Tokyo, Japan), and images were analyzed with Image-Pro Plus 6.0 (Media Cybernetics, Inc., Rockville, MD, USA).

### 2.10. Molecular Docking

The molecular docking was carried out according to previous reports and the methods are detailed in the Appendix A [23].

### 2.11. Data Statistics and Analysis

The data were presented as mean ± standard deviation (mean ± S.D.). Statistical analysis was performed using SPSS version 20, and multiple comparisons between groups were conducted using the LSD method (ANOVA). The significance level was set at *p* < 0.05 to determine statistical significance. The area of the wounds was determined using experimental images and analyzed by Image J software, which was developed by the National Institutes of Health in Bethesda, Bethesda, MD, USA.

## 3. Results

### 3.1. Analysis of CAPs Characterization Results

Molecular weights by LC-MS/MS detection were analyzed by searching the target protein database with the Peaks Studio program. A total of 762 peptides within the molecular weight range of 550–2300 Da were identified in the CAPs (Appendix A).

And, the main structures of the CAPs were analyzed by Fourier transform infrared spectroscopy (FTIR, Appendix A): Amide A bands (3299.00 cm^−1^ and 3410.35 cm^−1^) represents the presence of N–H bonds, amide B bands (2640.91 cm^−1^ and 2959.25 cm^−1^) represents the presence of C–H stretching, amide I bands (1658.78 cm^−1^ and 1621.66 cm^−1^) represents the presence of primary carbonyls (C=O), amide II bands (1396.12 cm^−1^ and 1404.68 cm^−1^) show CN stretching and NH bending, and amide III bands (1246.23 cm^−1^ and 1214.82 cm^−1^) show CN stretching vibrations with NH_2_ groups. The secondary structure of the CAPs was analyzed by FTIR to reveal its self-assembly effects, which showed that CAPs contain β-Sheet and other secondary structures [24,25], and was verified by the results of CD spectra (Appendix A).

The size of the CAPs increased with a self-assembly tendency as the concentration was raised from 50 mg/mL to 200 mg/mL, resulting in particles with sizes, respectively, ranging from 306.10 nm to 534.37 nm and 2132.00 nm to 5065.00 nm. Concurrently, the zeta potentials of the particles increased from −11.33 mV to −4.19 mV and from −11.20 mV to −1.97 mV. The self-assembly properties of the CAPs were analyzed by adjusting the solution pH and employing various techniques (Figure 1), including scanning electron microscopy (SEM), Zetasizer Nano ZS, CD spectroscopy, surface hydrophobicity (So), and atomic force microscopy (AFM). The findings indicated that the predominant secondary structure of the assembled CAPs at various pHs was β-Sheet, with a minor presence of a β-Turn structure. The self-assembly of CAPs was influenced differently by different pH values at two concentrations. The size of the particles for both low and high concentrations of CAPs was found to be the largest at pH 7.4 and pH 2.0, with all samples showing an initial increase in size followed by a decrease. The assembly behavior of the CAPs, which emerged mainly as a disc-shaped structure, was observed to be influenced by ionic strength and temperature, displaying reversible and pH-dependent microenvironmental characteristics (Appendix A). The results of this study provided a basis for further investigations into the self-assembly mechanism of CAPs.

LC-MS/MS analysis of 20 characteristic peptides of the CAPs revealed molecular weights ranging from 1162.55 to 2296.13 (amino acid residues 10–16) (Table 1). Notably, 47.86% of the total amino acids were hydrophobic (Ala, Val, Ile, Leu, Tyr, and Phe), while aromatic amino acids (Phe, Tyr, and Trp) accounted for 3.42% and 4.05%. These results indicate that CAPs are amphiphilic peptide-based polymers. Two fragments, Glu-Leu and Glu-Gly (Asp-Asn and Tyr-Glu-Leu-Pro-Asp-Gly-Gln), were found to recur within the 20 characteristic peptide sequences of the CAPs, so we speculate that it is these two fragments that play the main active functional role in CAPs. Furthermore, Val (Ala and Leu) appeared at the start of the characteristic peptide, while Lys and Thr (Lys) were found at the tail.

### 3.2. Verification System of In Vitro Healing Activity

#### 3.2.1. Test of Antioxidant Activity

The CAPs exhibited antioxidant activity (reduction ability power, Fe^2+^ chelating activity, and ABTS free radical scavenging ability) in vitro (Figure 2). The antioxidant activity of CAPs demonstrated a significant increase in a dose-dependent manner across all three indicators. Remarkably, the excellent antioxidant capacity of CAPs at 25 mg/mL is known from the result that the free radical scavenging capacity of ABTS is comparable to 5% of VC (Figure 2C).

#### 3.2.2. The Test of In Vitro Plasma Recalcification Time

The effect of CAPs on plasma recalcification coagulation time is shown in Figure 2E. The results demonstrate that both the CAPs’ group and the YN group had a significantly shorter plasma calcification time (*p* < 0.05) compared to the negative control group. A shorter plasma calcification time indicates a more favorable procoagulant effect. The results indicate that the CAPs have a certain degree of procoagulant effect.

#### 3.2.3. Cell Proliferation and Toxicity Test In Vitro

The fluorescence of acridine orange (AO) is caused by its ability to penetrate intact cell membranes, intercalate into nuclear DNA, and cause it to fluoresce bright green. Only damaged cell membranes can be penetrated by ethidium bromide (EB), which emits orange-red fluorescence. Apoptotic cells appear as staining enhancement, brighter, uniformly round or pyknotic, clumped structures. Nonapoptotic nuclei exhibit structure-like features with variable fluorescence. CAPs display distinct morphologies that are easily distinguishable. The results illustrated in Figure 2F indicate that cells cultured with CAPs on the surface of a Petri dish for 1, 2, and 3 days were stained green and did not show any red-dish-orange color, which suggests that the cells were viable. However, significant differences were observed between the groups. Furthermore, the reliability of the AO/EB double-staining results was confirmed using CCK-8, as demonstrated in Figure 2G.

The CCK-8 assay measures the color-depth and number of living cells, and there is a linear relationship between the two. This assay estimates the number of viable cells as dead or damaged cells lack dehydrogenase activity. Figure 2G shows that the different concentrations of CAPs had a more significant effect on promoting the proliferation of HaCaTs cells at 12, 24, and 48 h compared to the control group. At 12 h, a concentration of 12.5 μg/mL of CAPs and the positive control significantly promoted cell proliferation (116.43% and 121.00%, respectively) compared to other groups. At 24 h, 200 μg/mL of CAPs showed a proliferation-promoting ability close to that of the positive control group (*p* < 0.05).

In conclusion, based on the analysis of the results of AO/EB and CCK8, CAPs have some cytocompatibility and thus can be tentatively judged to accelerate wound healing by promoting cell proliferation.

### 3.3. Effects of CAPs on Animal Experiment

#### 3.3.1. The Effect on Wound Healing

We investigated the effect of the topical application of CAPs on wound healing in vivo by establishing a full-thickness mouse skin excision wound model. The wound area was measured on days 0, 2, 4, 6, 8, 10, 12, and 14 post-injuries, as shown in Figure 3B–D. Results indicated that CAPs significantly promoted wound healing compared to the control and YN groups. Figure 3C clearly indicates that the CAPs group exhibited the most effective healing effect among the three groups, followed by the YN group. It is noteworthy that between days 4 and 12, the healing rate in the CAPs group was significantly higher than in the other two groups.

#### 3.3.2. Effect of CAPs on Histopathological Analysis of Traumatic Surfaces in Mice

The impact of CAPs on skin wound healing in mice was investigated microscopically using H&E staining, as illustrated in Figure 3E. On the third day of the experiment, each treatment group displayed a lesser inflammatory response compared to the control group. On day 7, the YN group did not completely heal and in the control group, the dermis was sparse. Conversely, the CAPs group showed a coherent epidermis and granules in the dermis. This suggests that CAPs have stronger inflammatory inhibitory effects than the other two groups. After 14 days, there was no significant difference in the epidermis and dermis in the other groups compared to the control group, as observed in Figure 3B–D. Significantly, from all groups, the CAPs group showed the unique regenerative ability of hair follicle tissue in healing skin tissue. This distinguishes it from other groups and demonstrates its potential as a promising treatment for skin wound healing.

#### 3.3.3. Effect of CAPs on FGF and CD31 in Mice Wounds

Immunohistochemical analysis using CD31 allowed for specific labeling of vascular endothelial cells, indicated by black arrows in Figure 4A,B. Notably, there was a discernible difference in blood vessel formation among the six groups. The control group exhibited almost no new vessels, while the other groups exhibited a higher number of new blood vessels. Additionally, immunofluorescence analysis of FGF (Figure 4A) and quantitative results (Figure 4C) demonstrated that the CAPs group exhibited higher expression of FGF during the cell proliferation phase when compared to the control group. The expression level decreased gradually as the wound healing process entered the tissue remodeling stage.

#### 3.3.4. Effect of CAPs on Collagen and Wound Healing in Mice

In order to study the effect of the topical application of CAPs for reducing scarring, in one study, Sirius red picric acid staining was performed on mouse skin wound tissue on day fourteen. A visual comparison of the collagen I/III ratio in each group was performed and presented in Figure 5B, where the yellow or red fibers represent collagen I with a high birefringence and the green fibers represent collagen III with a low birefringence. The CAPs group exhibited significantly lower collagen I/III ratios compared to the control group (*p* < 0.05), as shown in Figure 5C. Figure 5D also plots the rate of scar reduction in mice at 14 days post-injury, which was consistent with the results seen in Figure 5A.

#### 3.3.5. Biosafety Assessment of CAPs In Vivo

After topical application, the spleen, liver and kidney from smear CAPs treatment groups were stained with H&E. Histology sections of the kidneys, heart, and liver were examined under a microscope. Following the administration of CAPs to smears, no apparent changes were observed in the histology (Figure 6). Therefore, it can be concluded that CAPs have no adverse effects on tissues, thus ensuring their safety.

## 4. Discussion

Skin wound healing remains a challenging issue despite extensive research efforts [26]. Peptide-based nanomaterials have gained wide acceptance due to their excellent stability and diverse structures compared to single peptides [27]. In this study, we developed nanomaterials composed of pH-responsive self-assembling CAPs, which provide a new approach for the controllable enzymatic enrichment of peptides and random self-assembly of peptides. To evaluate the in vitro healing activity of the nanomaterials, we established a verification system. Although large-sample statistics are necessary for reliable results, it is not currently feasible. Thus, we conducted comprehensive research to assess the precise biological activities and mechanisms of CAPs. Prior studies have revealed that in vitro activities such as antioxidant, antibacterial, procoagulant, as well as cellular proliferation are crucially related to the healing of soft tissue wounds.

The primary aim of this study is to investigate the self-assembly characteristics and mechanism of CAPs using a variety of characterization techniques. The particle size and potential of CAPs have been observed to vary in response to changes in pH, ions, and temperature (Appendix A); however, it remains unclear whether these changes are due to “self-assembly”, “self-aggregation”, or “aggregation”. Through FTIR and CD mapping, we have determined that CAPs are primarily composed of β-sheet and random coil structures and that self-assembly is the dominant mechanism. This has also been verified by SEM and AFM images, which reveal three primary morphologies of CAPs: disc-like, terraced nanodome-like, and granular nanostructure. These findings demonstrate that CAPs are peptide self-assembling materials and validate the use of controlled enzymatic digestion technology and pressurized ultrafiltration purification equipment for obtaining self-assembling peptides. Although the composition of the enzymatic digestion products remains diverse even after ultrafiltration purification, we have gained meaningful insight into the self-assembly mechanism through our physicochemical characterization results.

The physical and chemical analysis results, respectively, revealed that CAPs contain 762 and 618 different peptides (Table 1). This implies that CAPs not only provided various active peptides and nutrients to promote wound healing [8,28], but also contained a rich diversity of self-assembling peptide sequences and species. Notably, Glu-Leu is associated with antioxidation [29] and bone repair [30], Glu-Gly has a major impact on antioxidation [31] and anti-inflammatory [32], and Asp-Asn is crucial for antioxidation [33], vascular activity [34], and antibacterial [35]. Additionally, CAPs have an abundance of hydrophobic amino acids (such as Gly and Pro) and other amino acids (such as, His) that enhance their antioxidant activity (Figure 2). These findings provide a solid foundation for subsequent investigations of CAPs’ functional activities.

In summary, we have proposed a self-assembly mechanism for CAPs as illustrated in Figure 7. Amphiphilic peptides are the main components of CAPs, which form self-aggregates during ultrafiltration purification and rotary evaporation concentration based on the one-pot method principle (Figure 7) [16]. In practical applications, external factors such as solution pH and ion concentration induce the formation of peptide self-assembled materials. Under acidic pH, amino acid residues are protonated, electrostatic interactions are weakened, and the aggregation of amphiphilic peptide molecules is promoted by hydrogen bonding, π-π and hydrophobic interactions [36]. At alkaline pH, the reverse occurs, with additional electrostatic interactions inducing structural specificity of charged residues. Furthermore, π-π interactions of aromatic amino acids can induce directed growth [37]. The self-assembly of peptide fragments is facilitated by cations, such as Ca^2+^, Mn^2+^, and Zn^2+^, which promote the stacking and elongation of β-folds, resulting in the formation of higher-order aggregates with mixed structures. These aggregates may give rise to hierarchically oriented nanospheres, characterized by vertically stacked discs with a reduced diameter, which are stabilized by pi-stacking (Figure 1D and Appendix A) [38,39]. Self-assembled peptides generated as a result exhibit improved biocompatibility and degradability, setting CAPs apart from other covalent natural polymers [40].

Wound repairing is a complex process that requires careful management of both spatial and temporal transport of bioactive molecules, which can be achieved through the use of biomaterials [41]. In this regard, we conducted functional activity screening of CAPs using in vitro assays, including antioxidant, procoagulant and HaCaT assays, to provide a foundation for subsequent in vivo activity studies. The results indicated that CAPs exhibited good antioxidant and pro-proliferative effects, with some pro-coagulant effects and no cytotoxicity, consistent with the findings in Table 1 and Figure 1D. Since these functions are crucial for wound healing, we evaluated the pro-healing effects and mechanisms of topical application of CAPs using a full-thickness skin wound model. Our results showed that CAPs accelerated wound epithelialization, inhibited scar residue, and promoted hair follicle regeneration, meeting the functional requirements of all phases of skin wound healing. Specifically, CAPs inhibited the inflammatory response during the inflammatory phase (Figure 3E), promoted FGF cell proliferation and vascular regeneration during the cell proliferation phase (Figure 4), and facilitated collagen secretion, I/III ratio coordination, and nutrient delivery [42,43]. Most importantly, Figure 6 demonstrated the safety of the CAPs.

To gain a better understanding of the healing mechanisms of CAPs, we conducted molecular docking studies with EGFR1. Computational models play a critical role in predicting potential drug candidates in the drug discovery process [44]. Molecular docking simulations rank docked bioactive compounds based on their binding energies [45]. While CAPs have been shown to accelerate the process of wound epithelialization with no significant difference (Figure 3), the results of molecular docking reveal significant differences. Specifically, the caps-egfr1 docking was unsuccessful.

## 5. Conclusions

In this study, we extracted CAPs from *Crassostrea gigas* and found that they are composed of polypeptides with molecular weights of less than 3 kDa and have a characteristic sequence of supramolecular randomized self-assembly. Our in vitro experiments have demonstrated that CAPs possess antioxidant properties, promote coagulation, and enhance the proliferation of epidermal cells. These findings lead us to conduct in vivo experiments with a topical application of naturally safe CAPs, which effectively inhibited inflammatory responses, promoted revascularization, accelerated epithelialization, and balanced the I/III collagen composition ratio to prevent scar formation. Further research will explore the molecular mechanisms underlying the wound healing properties of CAPs.

## Figures and Tables

**Figure 1 antioxidants-12-01190-f001:**
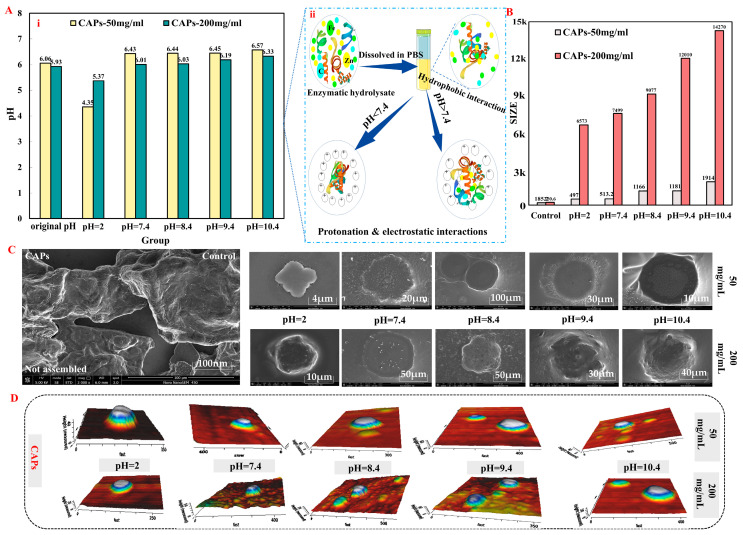
Characterization of self-assembly properties of CAPs. (**A**) The effect of different pH values in vitro on the pH values of CAPs. (**B**) Effects of pH value on averaged CAPs particle size. SEM (**C**) and AFM 3D (**D**) images of CAPs.

**Figure 2 antioxidants-12-01190-f002:**
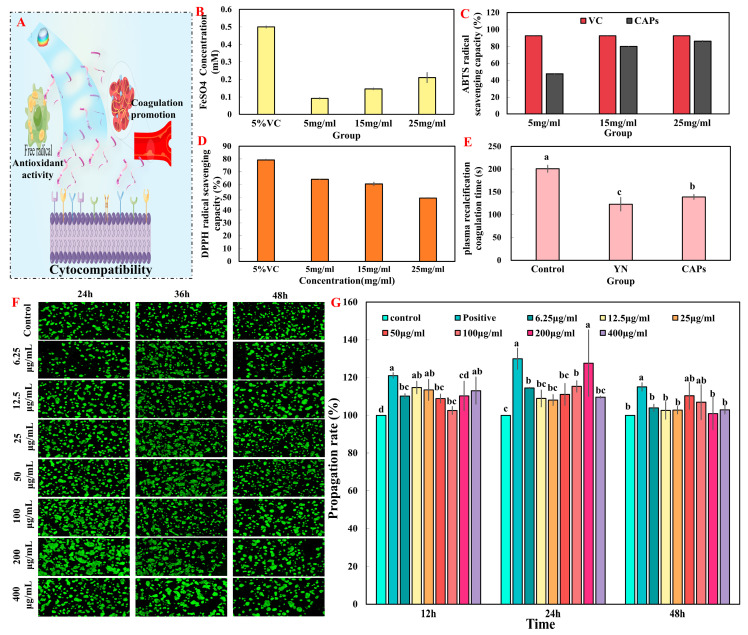
Validation of the in vitro activity of CAPS. (**A**) Conceptual diagram of the in vitro activity of CAPs. Reducing power (**B**) and ABTS radical scavenging activity of CAPs (**C**). (**D**) DPPH radical scavenging activity of CAPs. (**E**) Measurement of the coagulation time of CAPs in vitro. (**F**) AO/EB fluorescence dyeing with HaCaTs of CAPs for 24 h, 48 h and 72 h (magnification: ×5). (**G**) The CAPs viability of each group was determined by CCK-8 after incubation for 12, 24 and 48 h under specified conditions. The drug used in the positive condition was Human FGF-basic. Note: Superscript letters that are the same indicate that there is no significant difference (*p* > 0.05), whereas different superscript letters indicate a significant difference (*p* < 0.05).

**Figure 3 antioxidants-12-01190-f003:**
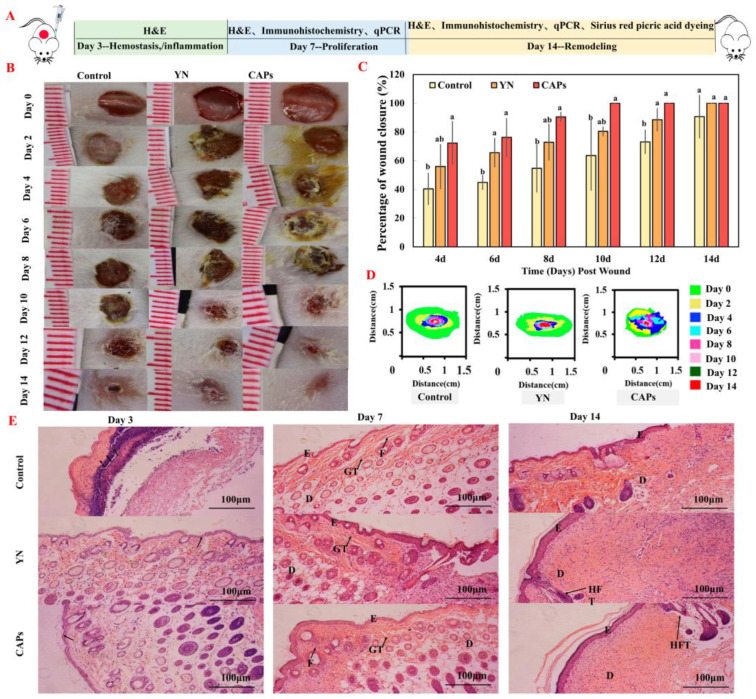
The effect of the CAPs on wound healing. (**A**) The experimental procedure of wound healing in mice. (**B**) Representative photographs of mouse wounds on days 0, 2, 4, 6, 8, 10, 12, and 14. (**C**) Percentage of wound closure for each group (measured every 2 days). Note: the use of the same letters indicates no significant difference (*p* > 0.01) within the same time period, while different letters indicate highly significant differences (*p* < 0.01). (**D**) Time-course of wound changes in mice over 14 days. (**E**) Histological analysis with H&E staining (10×). Note that black thick arrows indicate inflammatory cell infiltration. The abbreviations used in the images are as follows: D for dermis, E for epidermis, F for fibroblasts, GT for granulation tissue, and HFT for hair follicle tissue.

**Figure 4 antioxidants-12-01190-f004:**
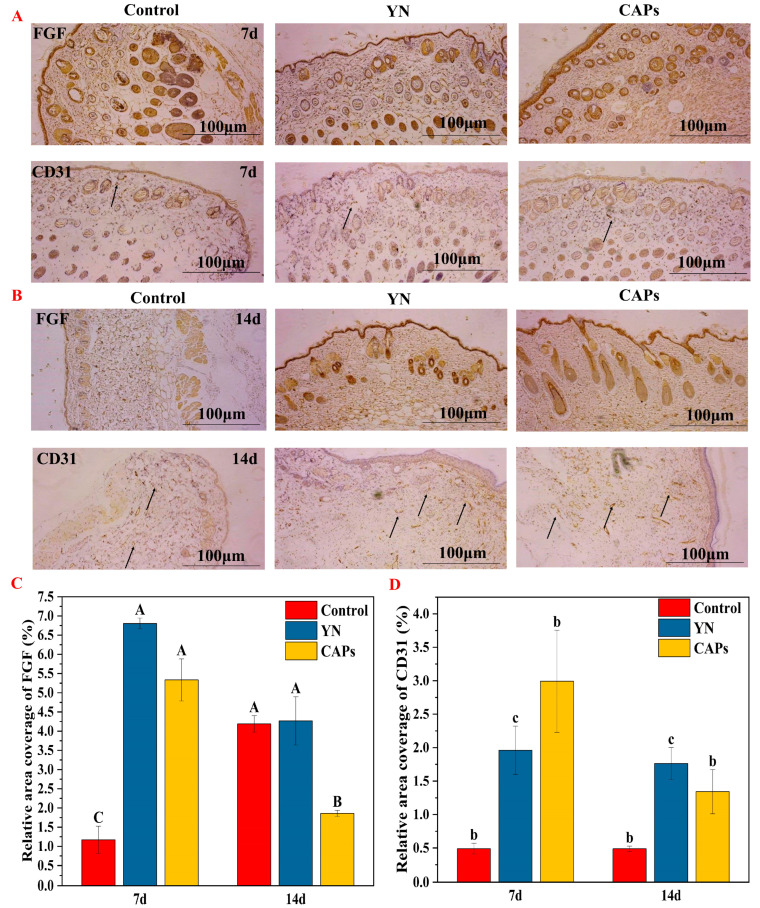
Using immunohistochemistry analysis of CAPs on mouse skin wounds. (**A**) Immunostaining images of FGF and CD31 in wounds from each group at day 7. (**B**) Immunostaining images of FGF and CD31 in wounds from each group at day 14. (**C**) The expression of FGF in wounds of each group was analyzed on days 7 and 14 after injury. (**D**) CD31 expression was assessed in wounds from each group at days 7 and 14 post-trauma. Note: the use of the same letters indicates no significant difference (*p* > 0.01) within the same time period, while different letters indicate highly significant differences (*p* < 0.01).

**Figure 5 antioxidants-12-01190-f005:**
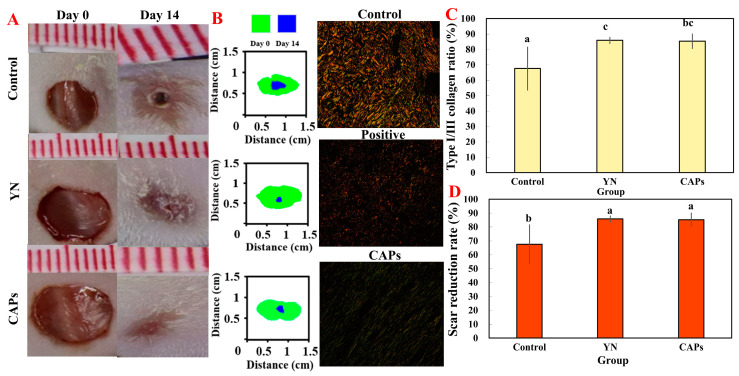
To assess the impact of applying CAPs on collagen synthesis in injured mouse skin, a study was conducted. (**A**) On day fourteen, the traces of the scar residue’s area were measured for each group. (**B**) Representative photo graphs shown demonstrate Sirius red stained collagen synthesis (magnification: ×5) where collagen III appears green and collagen I appears orange. (**C**) Collagen ratio in wound healing areas. (**D**) Based on the rate of scar reduction within each group. It is important to note that the same superscript letters indicate no significant difference (*p* > 0.05), whereas different superscript letters indicate significant differences (*p* < 0.05).

**Figure 6 antioxidants-12-01190-f006:**
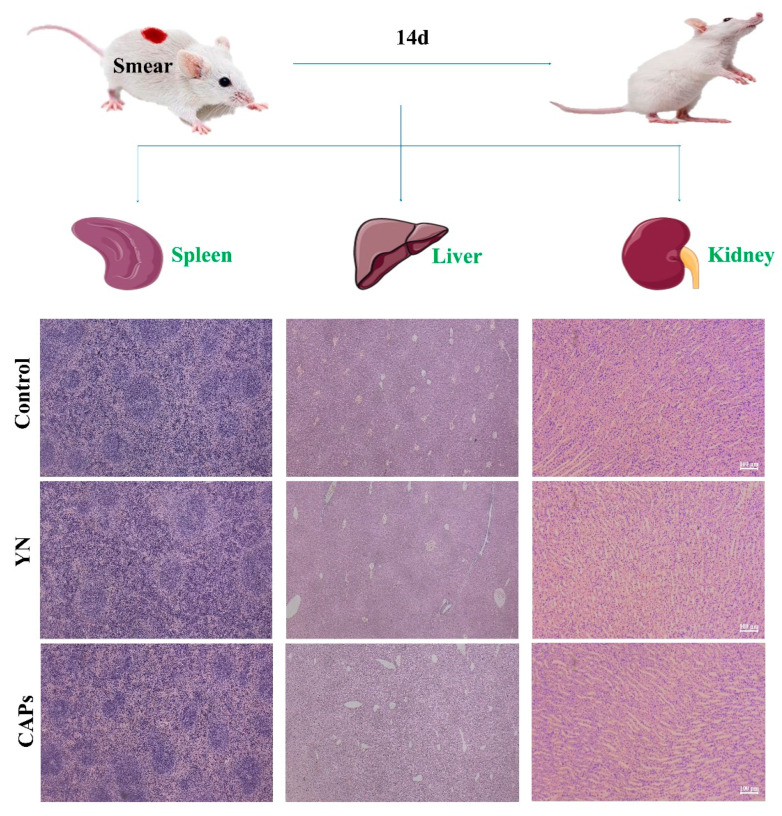
The in vivo biosafety evaluation of CAPs. H&E staining was used to assess the effects of CAPs on mouse spleen, liver, and kidney, both for the CAP group and for the YN group.

**Figure 7 antioxidants-12-01190-f007:**
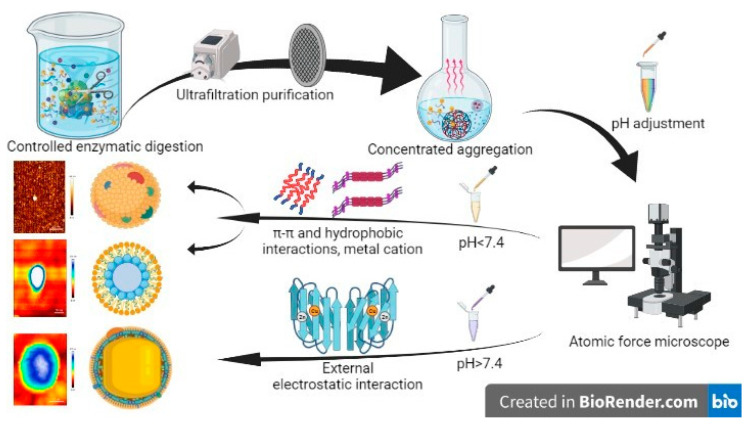
Self-assembly mechanism of CAPs.

**Table 1 antioxidants-12-01190-t001:** Main peptide sequences analysis of CAPs.

Sequence	Peptide Sequence of CAPs	Molecular Mass (Da)	Score
1	Leu-Gln-Glu-Lys-Glu-Glu-Glu-Phe-Asp-Asn-Thr-Arg-Arg-Asn-His-Gln	2071.967	52.05
2	Leu-Asp-Val-Asn-His-Asp-Gly-Lys-Ile-Ser-Ile-Glu-Asp-Val-Glu-Glu-Ser-Arg-Asn-Lys	2296.129	51.43
3	Leu-Asp-Glu-Leu-Glu-Asp-Asn-Leu-Glu-Arg-Glu-Lys-Lys	1629.821	48.84
4	Ile-Gln-Asp-Lys-Glu-Gly-Ile-Pro-Pro-Asp-Gln-Gln-Arg	1522.774	47.67
5	Asp-Glu-Leu-Glu-Asp-Asn-Leu-Glu-Arg-Glu-Lys-Lys	1516.737	45.66
6	Leu-Gln-Glu-Lys-Glu-Glu-Glu-Phe-Asp-Asn-Thr-Arg-Arg	1692.807	45.54
7	Leu-Glu-Lys-Ser-Tyr-Glu-Leu-Pro-Asp-Gly-Gln-Val-Ile-Thr	1590.814	45.34
8	Ile-Glu-Glu-Asp-Ala-Gly-Leu-Gly-Asn-Gly-Gly-Leu-Gly-Arg	1356.663	45.24
9	Leu-Arg-Glu-Lys-Asp-Glu-Glu-Ile-Asp-Ser-Ile-Arg-Lys-Ser-Ser	1803.933	44.6
10	Ile-Ser-Ile-Glu-Asp-Val-Glu-Glu-Ser-Arg-Asn-Lys	1417.705	44.16
11	Leu-Asp-Glu-Leu-Glu-Asp-Asn-Leu-Glu-Arg-Glu-Lys	1501.726	43.9
12	Ala-Ala-Asp-Glu-Ser-Glu-Arg-Asn-Arg-Lys-Val	1273.638	43.89
13	Asp-Val-Asn-His-Asp-Gly-Lys-Ile-Ser-Ile-Glu	1225.594	43.8
14	Leu-Arg-Glu-Lys-Asp-Glu-Glu-Ile-Asp-Ser-Ile-Arg-Lys-Ser	1716.901	43.56
15	Glu-Lys-Ser-Tyr-Glu-Leu-Pro-Asp-Gly-Gln	1164.53	43.46
16	His-Gly-Asp-Ser-Asp-Leu-Gln-Leu-Glu-Arg	1168.5472	43.42
17	Leu-Glu-Lys-Ser-Tyr-Glu-Leu-Pro-Asp-Gly-Gln-Val	1376.6824	43.15
18	Glu-Lys-Ser-Tyr-Glu-Leu-Pro-Asp-Gly-Gln-Val-Ile	1376.6824	43.05
19	Leu-Asp-Val-Asn-His-Asp-Gly-Lys-Ile-Ser-Ile-Glu	1338.6779	42.73
20	Ile-Thr-Gly-Glu-Ser-Gly-Ala-Gly-Lys-Thr-Glu-Asn	1162.5466	42.71

## Data Availability

Data supporting the reported results are contained within the article or are available from the corresponding author.

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
