# Peer review of "Marine-Derived Bioactive Peptides Self-Assembled Multifunctional Materials: Antioxidant and Wound Healing"

_antioxidants, 2023, doi:10.3390/antiox12061190_

Round 1

Reviewer 1 Report

The authors present an interesting topic, and it is clear that their area of expertise is in creating the peptide. Their main problem is that the wound healing studies are fairly weak. The section on creating the compound seems fine (although I have little expertise in that area). Here are the issues with the wound healing model:

1) Do you have approval from an Institutional Animal Care and Use Committee? Pentobarbital may not be an appropriate anesthetic. 

2) It appears that you leave the wounds open, so how do you keep the ointments in place?  In addition, dry wounds always take longer to heal than ones that maintain a moist environment. Simply, the ointment may in itself may improve healing, so just not treating the control wound may impair its healing. In addition, since the wounds develop debris and a "scab" covering the wound, how did you measure the edges of the wounds?

3) When measuring the wound, most investigators measure the area, not the diameter. Diameters are not typically uniform, so did you choose the largest diameter?  

4) FGF-10 is an unusual growth factor to look at in wound healing. It is involved with a lot embryologic development (ie. limb development). Typically, FGF-2 is examined as a key growth factor for angiogenesis. Why did you choose that growth factor? In addition, in the results, you call it simply "FGF". You should specify which FGF you are studying throughout the text since there are dozens of different types. In addition, your immunohistochemistry shows the greatest expression in hair follicles. I see very little other expression, but there are arrows pointing to something. Maybe greater magnification is needed. 

5) Please clarify your statistical methods. You say the "LSD" method. I have no idea what that means. Typically, ANOVA or a nonparametric statistic is used. 

Reviewer 2 Report

The work entitled “Marine-derived bioactive peptides self-assembled multifunctional materials: Antioxidant and wound healing” reports on the use of controllable enzymatic hydrolysis technology, using animal proteases, to obtain the supramolecular peptide self-assembling materials from the Pacific oyster. Data demonstrated the ability of the peptide-based material to respond to pH, has a pro-coagulant effect, free radical scavenging activity, and promote the proliferation of HaCaTs. Additionally, in vivo results demonstrated the potential of the peptide-based material to mitigate inflammation, boost fibroblast proliferation, and promote revascularization.

The subject is very well introduced; however, the novelty should be better emphasized. The methodology should have been completed. Forwarding researchers always to the supporting information file is not a good idea. The methodology should be provided in full leaving only to the supplementary file the additional data that is not required to the general comprehension of the methodology.

The discussion and presentation of the results is very comprehensive, supported by literature and to the point. Overall, the work is very well done. I would only recommend the previous alterations to be implemented prior to publication.

The english quality is fine. There are only small minor mistakes.

Reviewer 3 Report

Dear Editor, Dear Authors,

I was invited to evaluate the manuscript « Marine-derived bioactive peptides self-assembled multifunctional materials: Antioxidant and wound healing » by Dingyi Yu et al.

In this manuscript, the authors reporte about the use of materials formed by peptides from marine source with antioxidant and wound healing properties. The authors treated from Pacific oyster (Crassostrea gigas) by enzymatic hydrolysis technology, using animal proteases, to obtain the supramolecular peptide self-assembling materials (CAPs).  Authors demonstrated that the hydrolysis products consist of peptides ranging from 550 to 2300 Da in molecular weight, with peptide chain length was mainly 11-16 amino acids and are able to aggregate. Wound healing activity of those CAPs was explored in vitro and in vivo as well as their antioxidant effect showing free radical scavenging activity, and increase in the proliferation of human keratinocytes (HaCaT cells). Wound healing activity was confirmed in vivo with clear inhibition of inflammation, increased fibroblast proliferation and epithelialization process through revascularization. Authors concluded that their « CAPs can be regarded as a natural and secure treatment option with high efficacy in skin wound healing ».

The manuscript is interesting but I have however comments to be addressed.

1- Fig 2G : what was used as « positive » condition in Fig2G, please indicate it into figure’s legend.

2- Fog2G : I do not see here an effect on the cell proliferation. To demonstrate an effect on cell proliferation, the authors must seed the cells at low density at t0 and then measure the number of cells at t12h, 24, 48h and the cell number must increase over time for controls and potentially more with CAPs. The way the results are presented does not allow to see the proliferation as control are always at 100% at all time. And the t0 values are absent.

3- The authors mentioned they performed a wound healing test in vitro. Fig 2 is clearly not a wound healing test. Wound healing test is classically performed with a wound created with a tip and then measuring the closure of the wound over time. Please explain.

4- What about antimicrobial effect of the CAPs ? In a context of wound healing activity, someone will expect to see if the CAPs are also able to reduce/inhibit bacterial growth. Please perform such easy assay.

Regards

Round 2

Reviewer 1 Report

Accept 

Reviewer 3 Report

Dear Editor, Dear Authors,

Thanks to the authors that have addressed all my comments

regards